# Consumer and Carer Perspectives of a Zero Suicide Prevention Program: A Qualitative Study

**DOI:** 10.3390/ijerph182010634

**Published:** 2021-10-11

**Authors:** Victoria Ross, Sharna Mathieu, Jacinta Hawgood, Kathryn Turner, Nicolas J. C. Stapelberg, Matthew Welch, Angela Davies, Jerneja Sveticic, Sarah Walker, Kairi Kõlves

**Affiliations:** 1Australian Institute for Suicide Research and Prevention, School of Applied Psychology, Griffith University, 176 Messines Ridge Rd, Brisbane, QLD 4122, Australia; s.mathieu@griffith.edu.au (S.M.); jacinta.hawgood@griffith.edu.au (J.H.); k.kolves@griffith.edu.au (K.K.); 2Mental Health and Specialist Services, Gold Coast Hospital and Health Services, 1 Hospital Blvd, Southport, QLD 4215, Australia; kathryn.turner@health.qld.gov.au (K.T.); c.stapelberg@griffith.edu.au (N.J.C.S.); matthew.welch@health.qld.gov.au (M.W.); angela.davies@qmhc.qld.gov.au (A.D.); jerneja.sveticic@health.qld.gov.au (J.S.); sarah.walker@health.qld.gov.au (S.W.)

**Keywords:** zero suicide, suicide prevention, suicide attempt, patient care, mental health

## Abstract

This study explored the experiences of healthcare consumers who had recently attempted suicide, and their carers, following placement on a Suicide Prevention Pathway based on the Zero Suicide framework. Qualitative interviews were conducted with 10 consumers and 5 carers using a semi-structured interview schedule. Interviews were transcribed and thematic analysis was applied to identify prominent themes and sub-themes. Three interrelated themes were identified. The first theme was ‘Feeling safe and valued’ with the associated sub-theme pertaining to perceived stigmatizing treatment and self-stigma. The second was ‘Intersection of consumer and staff/organizational needs’ with a related sub-theme of time pressure and reduced self-disclosure. The final theme was ‘Importance of the ‘whole picture’, highlighting the relevance of assessing and addressing psychosocial factors when planning for consumer recovery. Overall, consumers and their carers reported a favorable experience of the Suicide Prevention Pathway; however, there were several areas identified for improvement. These included reconciling the time-pressures of a busy health service system, ensuring consumers and carers feel their psychosocial concerns are addressed, and ensuring that adequate rapport is developed. Key to this is ensuring consumers feel cared for and reducing perceptions of stigma.

## 1. Introduction

Suicide is a leading cause of death globally, with an estimated 703,000 deaths annually and up to 20 times as many episodes of suicide attempts and self-harm [1,2]. According to the Australian Bureau of Statistics (ABS), suicide was the 13th leading cause of death in Australia in 2019 [3]. Persons experiencing suicidal behaviors frequently present to hospital emergency departments [4,5]. The Gold Coast Hospital and Health Service (GCHHS) in Queensland (Australia) has one of the busiest emergency departments in the state. Between 2005 and 2015, 9045 persons (59.9% female) presented to the emergency department with non-fatal suicidal behaviors a total of 13,204 times (62.4% female) [6], with rates on the rise [7].

Guided by the principle that death by suicide is preventable, the National Action Alliance for Suicide Prevention developed the Zero Suicide framework of suicide prevention. The Zero Suicide framework provides a multilevel standardized structure for implementing evidence-based practices for suicide prevention in healthcare systems [8]. The framework relies on the systems of health care (generally hospital-based systems), rather than individual clinicians to prevent suicide, with the belief that every suicide is preventable within a ‘system’ [9]. Further, this system is reliant on a culture of no-blame and shared responsibility, collaborative safety, treatment, and recovery with a specific set of tools and skills for quality suicide prevention care and ongoing review and improvement of this care. The governing seven pillars of the Zero Suicide framework are: lead, train, identify, engage, treat, transition, and improve [8].

In 2016, the Gold Coast Mental Health Specialist Service (GCMHSS) initiated a Suicide Prevention Strategy based on the Zero Suicide Framework at the GCHHS—the first of its kind in Australia [10,11,12,13]. This model facilitates a positive cultural shift within the hospital setting and promotes better training and leadership among healthcare staff [10,14]. A key component of the GCMHSS Zero Suicide approach is the clinical pathway of care, the Suicide Prevention Pathway (SPP). The SPP is comprised of seven elements: screening, assessment, risk formulation, safety planning, preventing access to lethal means, structured follow-up, and transition, for at-risk individuals seeking care, and mandates the development and routine revision of a tailored safety plan in a collaborative, evolving process [10]. This pathway is designed to facilitate the widespread screening, assessment, follow-up, and importantly, transition of individuals seeking care back into the community. This is achieved through rapid, assertive follow-up upon discharge and co-developing plans for linking with ongoing supports and services in the community. This differs from traditional interventions which may focus on treating individuals for the medical consequences of their suicide attempt and discharging without community follow up.

All consumers presenting to GCMHSS following a suicide attempt are considered eligible for placement on SPP. Since its implementation in December 2016, over 7000 consumers at GCMHSS have been placed on the SPP. Emerging evidence indicates a range of positive organizational and clinical outcomes associated with high fidelity to the SPP model of care (specifically, a substantial increase in safety planning, addressing lethal means, and timely consumer follow-up in the community) and improved indicators of just culture [10,13]. Time-to-event analyses have also shown the SPP to reduce the risk of re-presenting with a suicide attempt by around 35% [15].

Further evaluation research is necessary to establish a strong evidence-base for the effectiveness of the ambitious and promising approach to suicide prevention program [9,16,17]. An essential and often underutilized aspect of health service evaluation and person-centered care is the inclusion of client-reported outcomes [18,19]. Therefore, the current study aimed to examine the experiences and perspectives of consumers who have been placed on the SPP, and their carers, to determine if their experience of the SPP was considered to be a satisfactory and effective process for aiding their recovery from suicidal thoughts and behaviors so as to inform future program refinements and implementation of the SPP.

## 2. Materials and Methods

### 2.1. Participants

The terms ‘consumers’ and ‘carers’ are used throughout the paper based upon the preferences of the participating health service. Consumers in this instance refers to healthcare patients who have experienced a recent suicide attempt that required intervention at the GCHHS, and carers refers to a close personal contact of the consumer involved in their intake and recovery. A purposeful sampling strategy using a maximum variation approach [20] was applied to ensure sample representation across gender, age, suicide methods, those who did/did not require inpatient treatment, and those who have/have not presented with repeated suicide attempts since being placed on the SPP. The approach was applied to ensure a wide range of experiences and perspectives from consumers. For inclusion, participants had to have progressed to the transition/warm handover stage of the pathway (i.e., discharge to community services). Exclusion criteria included cognitive impairment, intellectual disability, or diminished capacity to provide informed consent, as well as those who did not speak English.

The purposive sampling strategy involved the identification of two eligible groups of consumers (groups ‘A’ and ‘B’) matched on the same characteristics (i.e., gender, age group, attempt method, hospitalization, and re-presentation). Consumers from Group A were initially contacted, and for each individual unable/unwilling to participate, a person matched with the same characteristics from Group B was then contacted. This process continued with further consumers selected based on the characteristics not represented until the required sample was obtained and saturation was reached (i.e., no additional new information was attained).

A GCMHSS staff member assisted with identifying eligible participants and their carers via clinical records from the period July to December 2018. Those who agreed to participate and consented to be approached by the researchers were contacted by a clinical interviewer who arranged and conducted interviews in a private space provided by the GCMHSS. Participants were provided with written information, which included the identity and affiliation of the researchers, aims of the research, and confidentiality and informed consent issues, including the right to withdraw voluntarily. All participants provided written informed consent prior to taking part in interviews.

A total of three identified consumers declined participation, and a further three expressed interest, but due to scheduling could not attend the interview appointment. Half of the consumers also had a carer willing to participate. The final sample was 10 consumers (*n* = 6 male) aged 17 to 57 years (M = 36.8 years, SD = 17.27), and 5 carers (*n* = 2 male) aged 16 to 58 years (M = 34 years, SD = 18.43). Carers were comprised of a parent (*n* = 1), partners (*n* = 2), and daughters (*n* = 2).

### 2.2. Procedure

Fifteen individual face-to-face and telephone interviews were conducted by a female registered psychologist with expertise in suicidality and qualitative interviewing. Participants had no previously established relationship with the interviewer. A short introduction was provided to participants to explain the purpose of the study and interviews. The interviewer provided assurance to participants that support was available for them before, during, and after the interview.

A semi-structured interview schedule was developed to explore consumer-reported experiences at each key point along the SPP. Consumers were asked about their satisfaction with the SPP, aspects that did/did not work well, and how the SPP might be improved. A separate schedule was designed to gain carers’ perspectives. Some consumers chose to have their carers with them during the interview for emotional support, although all consumers and carer interviews were conducted separately. Interview duration was approximately 30–40 min for consumers and of slightly less duration for carers. This interview length may be seen as quite brief; however, this was in keeping with the practical and ethical considerations of interviewing someone who has recently experienced a suicide attempt (or cared for someone). Furthermore, despite their brevity, rich and informative data were still obtained. All interviews were recorded, de-identified, and transcribed verbatim for coding. The development of interview schedules, conducting of interviews, and data analysis were conducted independently from the GCMHSS. There were no repeat interviews or field notes taken, and due to time and resource constraints it was not feasible to provide participants with a copy of their transcript for further comment.

### 2.3. Ethical Approval

The study was approved by the Gold Coast Hospital and Health Service Human Research Ethics Committee (Ref: HREC/18/QGC/165. 06/08/2018). Research protocol and site-specific agreement approvals were also obtained from the GCMHSS.

### 2.4. Data Analysis

Using a constructivist grounded theory approach, a generic inductive thematic analysis was conducted where coding and theme development was directed by the content of the data, rather than fitting into a pre-existing coding frame [21,22]. Two authors (SM & VR) independently read the transcripts to familiarize themselves with the data and generated initial codes and themes. Using an iterative process, the researchers then worked with a third investigator (KK), ensuring the validity of analysis via on-going discussions and allowing for reassessment of themes and interpretations until consensus on the final themes and subthemes was reached. NVivo 12 Pro software (QSR International, Melbourne, Australia) was utilized for data management. The study was reported according to the 32-item Consolidated Criteria for Reporting Qualitative Health Research (COREQ) [23].

## 3. Results

Three dominant and interconnected themes were identified: Feeling safe and valued; Intersection of consumer and staff/organizational needs; and Importance of the ‘whole picture’. These themes encompassed each of the key points along the SPP.

### 3.1. Feeling Safe and Valued

There was broad agreement from consumers and carers that their experiences of the SPP were generally positive. This was particularly true in terms of consumers having a sense of emotional and physical safety, whereby consumers and carers described the importance of physical comfort as well as emotional security, feeling understood, and having their needs assessed and cared for. Mutual respect in interactions with staff was highlighted as crucial, with carers noting the importance of staff keeping them informed. Overall, consumers and carers acknowledged that staff were mostly empathetic and caring. These experiences extended across both their hospital stays and their engagement with mental health staff in the community during the transition and follow-up stages of the pathway. Those consumers who felt safe and valued also reported they felt listened to and were able to share their ‘story’.
*I felt really safe in there. I felt comfortable with the nurses—they always looked after me and made sure if I wanted something to eat or if I was comfortable or if I was too cold they would grab me a blanket. So, they were very respectful and just made sure I was okay at all times.*(Female consumer: experience with hospital staff)
*She was pretty respectful. Caring in a general sort of a way. I mean, it wasn’t overly gushy, pat on the back and look after you. It wasn’t touchy-feely, but there was a—there was a respect there which was quite good.*(Male consumer: experience with community follow up)

#### Stigma and Shame

A sub-theme of stigma and shame was also identified by consumers. Although most participants indicated they felt some level of emotional safety and respect, others reported that this was not always experienced, highlighting the complexity of emotions and varied experiences of suicidal consumers. For many consumers, it was clear that their stigma and shame was self-directed. Several consumers described feeling embarrassed and ashamed of their actions and feeling like a burden to health professionals and to their own families.

However, some consumers also perceived their experiences and treatment from staff as stigmatizing and alienating. For example, several consumers believed staff made comments that conveyed a sense of ‘pre-judgment’. Furthermore, over half of the consumers interviewed described their hospital experiences using prison analogies or metaphors (e.g., ‘it felt like you were in prison’, ‘segregation’, or referred to their discharge as being ‘released’). This was compounded by the vulnerability of these consumers, many of whom had impaired memories of their hospital experience, were under the influence of substances, and/or were disoriented upon admission. Perceived stigma was counterproductive in establishing a rapport with staff and in some situations was reported to contribute to consumers adopting an uncooperative attitude toward staff members, which led to further breakdowns in communication.
*One of the nurses was a bit abrupt with me, so I think she was like ‘oh, another nutcase has come in’ you know? She wasn’t very warm and fuzzy.*(Female consumer: experience upon waking in hospital)
*That’s what you’re made to feel it is [a crime] and that’s one of the drawbacks… you’re just coming around, you’re coming to terms with you’re actually still alive, which is a shock, because you don’t want to be. You go from that transition to a stark cell; you’re being punished for a crime that you didn’t commit…That’s where prejudging, making such a quick, snap judgement about somebody is also wrong.*(Male consumer: experience of mental health staff during observation)

### 3.2. Intersection of Consumer and Staff/Organisational Needs

Many consumers believed that staff priorities/organizational procedures impeded their ability to connect with and respond to consumer needs. One key and complex interplay that manifested across several interviews was the intersection between staff safety (a paramount organizational priority) and the consumer’s emotional needs (a crucial aspect of safe and consumer-centered care). This was reported at each stage of the SPP. For example, discomfort and surprise at the number of different paired workers arriving at their homes for follow-up; or the physically uncomfortable and isolated assessment rooms used for consumers who were assessed as a danger to themselves or others in the emergency department. Despite this, consumers and carers described how they understood the various pressures and competing interests for staff and generally acknowledged that staff were ‘trying their best’.
*I kind of put it down to—you know, they’ve got a job to do at the end of the day and they probably see this kind of stuff on multiple levels kind of thing. But at the same time, I kind of felt as if it was just kind of just like—oh another one of ‘these people’.*(Male consumer: experience with hospital staff)
*I was just a bit uncomfortable how many people came out each time to see me. I had four different appointments with four different paired workers came out each time and it was just uncomfortable because I had to introduce myself again and I was telling them pretty much what I’m going through.*(Female consumer: experience with community follow up)

#### Taking Time and Reduced Self-Disclosure

Whilst the importance of establishing rapport and building trust was identified by both consumers and carers, this was impacted by the time-demanding nature of mental health services, combined with the perceived stigma and shame described previously. Many consumers and carers perceived staff members to be rushed, or that staff were ‘just doing their job’ and merely ‘ticking boxes’ to get through paperwork to move consumers through the system. Some participants reported this as a barrier to their willingness to disclose their true needs or feelings. This sub-theme also highlighted a push-pull dynamic between consumers and staff. On the one hand, consumers who felt judged or who did not have time to build rapport were withholding information, yet on the other hand, they acknowledged help was there for them should they be accepting of it. Alternatively, some consumers expected staff to probe deeper or be more understanding despite the limited self-disclosure of the consumer and/or being time-poor themselves. Other consumers withheld important information for fear of re-hospitalization or having their means of suicide restricted or taken away.

Balancing the demands of the organization and the needs of the consumers was a key recommendation made by consumers and carers (including those who recounted positive interactions). In particular, consumers and carers felt it was important to spend more time with participants at key stages of the SPP.
*He was sort of just like every other psychiatrist, trying to get through each patient as quickly as you can and move on to the next. There’s a lot of people that they’ve got to get through and unfortunately that’s just how it is sometimes, which I understand.*(Female consumer: experience with inpatient mental health assessment)
*They were definitely caring, the psychiatrists that came. They were definitely… if I had said I need help I know they would have given me help, but I just wanted them to go away at that stage.*(Female consumer: experience with inpatient mental health assessment)

### 3.3. Importance of the ‘Whole Picture’

A further interrelated theme emerged whereby psychosocial factors were highlighted by consumers and carers as important in the lead up to their suicidal experience and in their experience of the SPP across all stages. Over half of consumers cited a wide range of contextual factors as integral motivations for their suicide experience (e.g., financial difficulties, breakdown in relationships, and perceptions of inadequate social service support). It was reported that many of these stressors went on to impact their recovery. Having the opportunity and feeling able to share these factors and their ‘story’ during the screening/assessment stages was considered beneficial to those consumers who were able to do so, and a missing element for those who could not.
*They assumed straightaway it was a spontaneous act. They didn’t ask about any of the lead-up or the triggers or anything else. At some point in the conversation, I explained that a lot of this is frustrations through my medical history, through the system with Centrelink that forces you into poverty.*(Male consumer: experience with mental health assessment)
*Maybe to just have people that would listen to you and actually ask you questions, ask you why you’re there and why this happened, not just how, and actually care I guess… I feel like I probably would have felt better faster, I probably would have got this big, heavy weight lifted off of me [had the consumer been able to share their story].*(Male consumer: experience with hospital staff)

#### Addressing Psychosocial Needs

Psychosocial factors and supports were also considered to be important in the safety and care planning stages of the SPP. A small proportion of consumers claimed that they did not leave hospital with a safety plan; a mandated component of the SPP. Half of the consumers interviewed reported their plan was not useful/tailored and could have benefited from more recognition of social determinants (e.g., access to transport for service appointments or discomfort calling helplines). Some carers noted that they were not included in the planning yet named in the document. Many consumers and carers highlighted the difficulties of completing a safety plan, particularly prior to discharge where they described an urge to complete it quickly and go home, impacting their engagement with the process. Consumers conveyed their appreciation of the Acute Care Team reviewing the safety plan during the at-home follow up, and in particular anticipating that the pressured hospital environment may mean the plan no longer ‘resonated’ or was impacted by impending discharge. The development of a care plan at this stage of follow-up that specifically targeted psychosocial drivers was considered beneficial. Furthermore, for those consumers who identified with their individually tailored safety plan that acknowledged their psychosocial needs and preferences, this was viewed as extremely positive.
*… the spectrum of problems that I’ve got going on, financial, it’s situational, it’s I don’t have a license to be able to move if I wanted to, I can’t feel like I can get work without a license, apart from call centers, which affect my mental health. I’m paying $320 a week rent on $660 a fortnight income. There’s a lot of different aspects, apart from the other things that brought me to it.*(Male consumer: experience with safety planning)
*You feel like you’re rushed when you’re doing it. So, it’s discharge day, ‘okay let’s get the safety plan out, okay let’s add to it’, so it was more like a tick the box exercise. I think that [would be] a great idea, to do it earlier in the admission.*(Female carer: experience with safety planning)

## 4. Discussion

This study applied a qualitative approach to examine consumer and carer experiences of the GCMHSS SPP. To the best of our knowledge, this study is the first to explore consumer and carer experiences as part of a Zero Suicide Framework, in Australia or internationally.

From consumer and carer perspectives, feeling valued, respected and emotionally safe in interactions with healthcare staff was key. This is supported by previous research which highlights the importance of empathy and a strong therapeutic alliance as key markers of positive experiences of mental health services by those experiencing suicidal crisis [24]. Furthermore, person-centered, empathetic, and respectful treatment is a central tenant identified within Delphi-derived expert guidelines for the care and follow-up of suicidal individuals [25,26], and interpersonal trust/social capital has been shown to be protective of suicidal thoughts (e.g., [27]). Consumer and carer accounts obtained in this study supported the value SPP places on rapport and trust in building a relationship between clinicians and consumers, many of whom may not have previously felt safe enough to disclose their emotional pain or circumstances preceding their suicide attempt.

Given the importance of feeling valued and emotionally safe, it is unsurprising that the inverse of such experiences was described as particularly difficult. Consumers’ self-stigma and perceptions of stigmatizing treatment by staff contributed to a breakdown in relationship and hindered assessment, safety planning and consumer engagement. Such attitudes and stigmatization exacerbate poor self-esteem and increased hopelessness [28], both of which are further implicated in the experience of suicidality [29,30]. Stigma/shame also increases secrecy and reduces the likelihood of help-seeking [31] and is described as a defining feature of negative experiences of mental health care [24]. In our study, internalizing negative stereotypes and using self-stigmatizing language was common, with some consumers describing themselves as a burden and embarrassed by their hospitalization. Anticipated or perceived stigma serves as a self-defeating process in terms of seeking help and/or experiencing health gains and is an important consideration in prominent models of (mental) health stigma [32,33]. A previous qualitative study that investigated the information needs of those who attempted suicide, found that information to challenge stigma and address negative attitudes is key to recovery, and that practical information that supports understandings around recovery, as well as information that induce hope, should be included in the post discharge process [34]. Such information may be emphasized during the assessment, safety planning, or transition phase of the SPP.

A challenging interplay was identified between the expectations/needs of the consumer and their carers, and the priorities/needs of the organization and staff. Whilst spending the appropriate amount of time with a consumer in crisis may be logistically difficult when faced with the competing demands of a busy emergency department, it remained an important keystone of suicide intervention for these consumers and their loved ones. To expedite patient movement from the emergency department to inpatient units or discharge, the GCHSS must comply with the National Emergency Access Target (NEAT) 4 h discharge rule where medical intervention and mental health assessment must occur within the specified timeframe. While this may facilitate better consumer flow, it often does not translate to positive perception of patient care [35]. Allowing consumers additional time to feel understood may help with disclosure of suicidal thoughts and intent and may also go toward reducing the feelings of disorientation upon admission and discharge reported by consumers [36].

The final theme highlighted the importance of diverse psychosocial factors, such as financial issues and relationship breakdowns, in contributing to suicidality. These factors, such as financial difficulties and unemployment, consistently emerge within the literature as important in predicting suicidality (e.g., [37,38]). Some consumers felt that their safety planning was not tailored to meet their individual psychosocial needs, and subsequently questioned the utility of the plan. Currently, there is limited systematic evaluation of suicide safety planning as an intervention [39,40,41,42]. Nevertheless, expert guidelines stress the importance of a tailored and collaborative discharge care plan that highlights relevant strategies for the consumer and provides them with autonomy over ongoing treatment [25]. Although the precise objective of safety planning is not to respond to psycho-social needs or determinants of suicidality, such as relationship, financial, socio-legal, or physical/mental health problems [43], some consumers in the current study believed the process should include and respond to these factors. Similar findings were obtained in a study investigating the support needs of people after a suicide attempt in South Africa [44], where participants identified a need for support in dealing with interpersonal conflict and solving socio-economic problems. Addressing these needs occurs in the transition phase of the pathway during the structured follow up sessions, where tailored referrals to services within the community and practical support are intended to alleviate associated psychosocial distress contributing to suicidality (a key component of the care plan). Of note, consumers reported benefits of the routine revision and evolving safety planning that is facilitated during these follow up visits, which address some of the challenges in the initial Safety Planning within the environment of a busy emergency department or upon discharge from an inpatient stay. Consumers may benefit from enhanced psychoeducation that explains the rationale of safety planning (as a tool for risk management) and care planning (that addresses the psychosocial drivers of suicidality) as distinct steps of the SPP.

The results of this study indicate that consumer and carer experiences of the GCMHSS SPP were largely positive: they felt safe and were appreciative of the care and follow-up engagement they received, and these findings suggest the pathway was a satisfactory experience for aiding their recovery. However, the study also identified areas for improvement. Our results are consistent with recent research indicating that reducing stigma, comprehensive psychosocial assessment, and addressing structural barriers to accessing care are essential recommendations for improving mental health services for those who have attempted suicide [45]. As part of several actions to improve the pathway, the GCMHSS has recently introduced a new Crisis Stabilization Unit as an alternative to the emergency department for suicidal consumers. This initiative will have a strong involvement of peer workers with lived experience, and will not be subject to the 4 h discharge rule which should address the issue of consumers feeling rushed and needing more time with staff to build a rapport and trust [46]. The GCMHSS are also focusing on improving inpatient staff training. This approach will be critical in facilitating improvements for inpatients, for example by ensuring safety planning is conducted much earlier in the pathway, rather than on the day of discharge. Nevertheless, there exist several practical issues that impede the resolution of staff versus consumer priorities identified in this study, such as under-funding of mental health services in Australia and associated burnout in mental health professionals (e.g., [47]) and the ‘burden’ of excessive and often redundant paperwork at the expense of efficiency and consumer engagement (e.g., [48]).

### Strengths, Limitations and Future Research

A strength of this study was the consumer-centered and qualitative approach to obtaining an in-depth understanding of consumers’ perceptions. These insights are critical for informing future developments and improvements to the SPP. A limitation, however, is that the qualitative design and small sample size limit the generalizability of findings. For instance, more detailed demographic and contextual information regarding consumers and their carers was not collected (e.g., ethnicity, migrant/refugee background, and mental health problems). Given that not all people who attempt suicide present to hospital for medical intervention, and that some social groups may be even less likely to present due to systemic or cultural reasons, it is crucial that future research examines who is and who is not engaging with services. Nevertheless, the use of a qualitative design enabled the collection of rich data directly from the perspectives of consumers and carers that can be used to generate future research and improve experiences of healthcare-based interventions for a suicide attempt. The application and integration of quantitative or mixed methods studies is recommended for future evaluation research. It was also difficult to discern from participants’ accounts whether they were referring to emergency department staff or mental health staff within these departments and thus their quotes were described in general terms (e.g., regarding ‘experience with hospital staff’). This is likely due to lack of consumer/carer knowledge of hospital systems and staffing and may suggest the need for clarification to aid consumer navigation of complex health systems. Given the interplay between organizational priorities and consumer needs/expectations, future research may also benefit from exploring staff experiences with the SPP as this will facilitate better understanding of its challenges and ultimately improve consumer experiences in the future. Based on the current findings, a further and important consideration for future research would be examining the impact of the implementation of the Zero Suicide Framework and a refined SPP within the GCMHSS over the course of the COVID-19 pandemic. International studies have not shown a substantial increase in suicides [49,50] or self-harm [51] in the initial timeframe of the pandemic. However, this may be due to the mitigating effects of governmental economic supports that are not likely to continue indefinitely [52]. Given the importance of economic and psychosocial factors to consumers in the current study conducted prior to the pandemic, and the ongoing uncertainty of the impact of COVID-19 on suicidal behaviors, this is an important area of ongoing and future research.

## 5. Conclusions

Findings revealed that consumers and their carers generally described their experiences as positive, although there were still areas identified for improvement. These included ensuring consumers feel listened to, addressing perceptions of stigma, more consideration of psychosocial circumstances, and ensuring adequate rapport is developed. A particularly challenging barrier was the need to balance the time-pressures of a busy health service system and consumer preferences for spending time with their clinicians, building rapport and feeling heard. These findings are consistent with previous research and provide important client-centered indications for improving Zero Suicide service delivery when providing appropriate healthcare to those who have experienced a suicide attempt.

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
