# Peer review of "Consumer and Carer Perspectives of a Zero Suicide Prevention Program: A Qualitative Study"

_ijerph, 2021, doi:10.3390/ijerph182010634_

Round 1

Reviewer 1 Report

Thank you for giving me the opportunity to review this interesting and meaningful piece of work. Below a few suggestionsç

  1. Page 1 link 35 there is a typo, it should be between
  2. As the aim of the paper is to evaluate the intervention, it would be useful to provide more information in the introduction about the framework and the specific information and highlight how it differ from other hospital based clinical interventions.
  3. The analysis process described does not seem to follow grounded theory informed Thematic analysis (see for example https://www.psychologicalscience.org/observer/developing-theory-with-the-grounded-theory-approach-and-thematic-analysis) but a classical TA approach, please clarify
  4. The limitations should emphasise the significance of integrating qual and quant evaluations rather than suggesting quant measures as they have their own limitations when used on their own. One of the main limitation of this study is that there is no consideration of the contextual aspects of suicide prevention. All we know is the age and gender of the participants but we know that some part of the population like people of indigenous, migrant and refugee backgrounds tend to use hospital and generally clinical services for mental health issues at a lower degree. If this info was not collected, the authors should at least mention that it is more likely that this intervention might be used by some sectors of the population and future studies should explore who is using and who is not using this intervention.

A part from the issues raised above, I believe this was a well thought and useful piece of work and look forward to see it published, thank you!

Author Response

Reviewer 1

Thank you for giving me the opportunity to review this interesting and meaningful piece of work. Below a few suggestions:

  1. Page 1 link 35 there is a typo, it should be between

Thank you – this has been corrected.

  1. As the aim of the paper is to evaluate the intervention, it would be useful to provide more information in the introduction about the framework and the specific information and highlight how it differ from other hospital based clinical interventions.

This is an important point and in conjunction with suggestions made by reviewer 2 we have now added additional information to the introduction about the suicide prevention pathway and described how this differs to other hospital-based interventions. “This pathway is designed to facilitate the widespread screening, assessment, follow-up, and importantly, transition of individuals seeking care back into the community and their families. This is achieved through assertive follow-up upon discharge and co-developing plans for linking with ongoing supports and services in the community. This differs from traditional interventions which may focus on treating individuals for the medical consequences of their suicide attempt and discharging without community follow up.”

  1. The analysis process described does not seem to follow grounded theory informed Thematic analysis (see for example https://www.psychologicalscience.org/observer/developing-theory-with-the-grounded-theory-approach-and-thematic-analysis) but a classical TA approach, please clarify

We thank the reviewer for this comment. In this context, we have applied a “Constructivist” Grounded Theory approach, a research method that focuses on generating new theories through inductive analysis of the data gathered from participants rather than from pre-existing theoretical frameworks. We believe this is an appropriate approach for conducting the thematic analysis in the context of this research.

  1. The limitations should emphasise the significance of integrating qual and quant evaluations rather than suggesting quant measures as they have their own limitations when used on their own. One of the main limitation of this study is that there is no consideration of the contextual aspects of suicide prevention. All we know is the age and gender of the participants but we know that some part of the population like people of indigenous, migrant and refugee backgrounds tend to use hospital and generally clinical services for mental health issues at a lower degree. If this info was not collected, the authors should at least mention that it is more likely that this intervention might be used by some sectors of the population and future studies should explore who is using and who is not using this intervention.

These are important considerations, and we thank the reviewer for drawing attention to our need for inclusion in the limitations and future research section. In response we now have the following few sentences (added material in bold) “A limitation, however, is that the qualitative design and small sample size limit the depth and generalizability of findings. For instance, more detailed demographic and contextual information regarding consumers and their carers was not collected (e.g., ethnicity, migrant/refugee background, mental health problems). Given that not all people who attempt suicide present to hospital for medical intervention, and that some social groups may be even less likely to present due to systemic or cultural reasons, it is crucial that future research examines who is and who is not engaging with services. The application and integration of quantitative or mixed methods studies is recommended for future evaluation research.”

Apart from the issues raised above, I believe this was a well thought and useful piece of work and look forward to see it published, thank you!

Reviewer 2 Report

Many thanks to have the opportunity to revise this manuscript aiming to explored the experiences of healthcare consumers who had recently attempted suicide and their carers following placement on a Suicide Prevention Pathway based on the Zero Suicide framework.

As underlined by the authors, the prevention of suicide is one of the most important and significant strategy to improve the outcome of subjects at risk and their carers. The study is interesting but it is conducted on a very small sample (10 consumers and 5 carers). This is the major limitation, and the findings can not be generalized.

Furthermore, the program prevention was not described but the authors write only that consists of seven phases of interview and last 30-40 minutes. 

In my opinion, this manuscript should be presented as letter to editor, and not as a clinical article. Unfortunately, i regret to inform you that this manuscript can not processed in this original presentation. 

Author Response

Review 2

Comments and Suggestions for Authors

Many thanks to have the opportunity to revise this manuscript aiming to explored the experiences of healthcare consumers who had recently attempted suicide and their carers following placement on a Suicide Prevention Pathway based on the Zero Suicide framework.

As underlined by the authors, the prevention of suicide is one of the most important and significant strategy to improve the outcome of subjects at risk and their carers.

The study is interesting but it is conducted on a very small sample (10 consumers and 5 carers). This is the major limitation, and the findings can not be generalized.

We thank the second reviewer for their recognition of the importance of suicide prevention and improving outcomes for people who have recently attempted suicide and those who are caring for them in the community. We agree with the important point that the study was conducted with a “small” sample; however, it did meet the parameters of data saturation befitting the qualitative design. We have also reported on our sampling strategy that aimed to maximise variation across demographic characteristics in this hard to reach and vulnerable population. We note this on page 3. We note that this “small” sample size could be seen as a limitation regarding generalizability of findings and have now added a further justification for future research -“Nevertheless, the use of a qualitative design enabled the collection of rich data directly from the perspectives of consumers and carers that can be used to generate future research and improve experiences of healthcare-based interventions for a suicide attempt.”

Vasileiou, K., Barnett, J., Thorpe, S. et al. Characterising and justifying sample size sufficiency in interview-based studies: systematic analysis of qualitative health research over a 15-year period. BMC Med Res Methodol 18, 148 (2018). https://doi.org/10.1186/s12874-018-0594-7

Furthermore, the program prevention was not described but the authors write only that consists of seven phases of interview and last 30-40 minutes.

Thank you for raising this important point. This sentiment was shared by the first reviewer. We do briefly describe the suicide prevention pathway which is comprised of seven elements in the introduction. In response to the reviewer’s comment, and that provided by reviewer 1, we have now elaborated on this description and provide the following “This pathway is designed to facilitate the widespread screening, assessment, follow-up, and importantly, transition of individuals seeking care back into the community and their families. This is achieved through assertive follow-up upon discharge and co-developing plans for linking with ongoing supports and services in the community.” Regarding the qualitative research interview lasting 30-40 minutes we have further elaborated that this may be seen as quite brief however was in keeping with a responsive and respectful approach to vulnerable individuals who have recently attempted suicide, and that rich, informative data was still obtained.

In my opinion, this manuscript should be presented as letter to editor, and not as a clinical article. Unfortunately, I regret to inform you that this manuscript cannot processed in this original presentation. 

As far as we are aware there is no option for a ‘letter to the editor’ format of article for submission the International Journal of Environmental Research and Public Health. We believe the qualitative design is an important strength and was conducted in line with international guidelines Page 3 (line 147). Findings provide important information as to the experience of patients who have recently attempted suicide, and their carers, in navigating the hospital-based healthcare system and their perspectives on a comprehensive, assertive suicide prevention program designed for these individuals. Nevertheless, we would be happy to consider a ‘short communication’ if still preferred by the Editor and reviewers.

Reviewer 3 Report

Thank you,

Best regards

Author Response

Review 3

Thank you for the opportunity to review this really interesting paper.
Overall:
- The paper addresses a highly topical issue.
- Practical implications, relevant to the clinical reader, are evident.
- I personally appreciated the originality of the paper, the important practical implication
translated into a pragmatic framework, and the attention to the personal experience of patients.
- However, the paper can be improved through minor revisions. They mainly concern the
enrichment of the references, in order to make explicit the cultural background and to
strengthen the generalization to a wide audience of the important results.

We thank the reviewer for their careful and constructive feedback and are encouraged by their positive reception to our work. We appreciate the opportunity to strengthen the manuscript.

As for the main specific sections:
- The Introduction incisively introduces the issues that the review will develop next, and it is
clearly organized. Observation: Authors are encouraged to unify the formulation of "aim" and
objective" at the end of the introduction.

Thank you for this valuable feedback. We have now unified the aim and objective at the end of the introduction “Therefore, the current study aimed to examine the experiences and perspectives of consumers who have been placed on the SPP, and their carers, to determine if their experience of the SPP was considered to be a satisfactory and effective process for aiding their recovery from suicidal thoughts and behaviors so as to inform future program refinements and implementation of the SPP.”

- The Methods are thoroughly and appear rigorous. The explicit referral to grounded theory is
very appropriate. I do not believe that the small number of subjects involved affects the results (although that subject is rightly highlighted in the limitations), as this is a qualitative study. The authors stated that there was informed consent for participants and approval from the Ethics Committee, so there are no ethical concerns.

Thank you.

- The Results are fully exposed.

Thank you.

- The Discussion is well written, coherently with the aim of the work; it can be enriched by
some references that the authors may consider.

We thank the reviewer for their positive feedback and their time and consideration in providing these additional references that can be used to enrich our current work. We have read with much interest and enthusiasm the below references and address each of the reviewers numbered points in turn.

1) About stigma, some general (but widely used in the literature) references are: “Stigma,
prejudice, discrimination and health” Soc Sci Med 2008; 67(3): 351,
doi:10.1016/j.socscimed.2008.03.023; “The Health Stigma and Discrimination Framework: a
global, crosscutting framework to inform research, intervention development, and policy on
health-related stigmas” BMC Med 2019; 17(1): 31, doi:10.1186/s12916-019-1271-3, and the
pivotal paper of Thornicroft “Global pattern of experienced and anticipated discrimination
against people with schizophrenia: a cross-sectional survey” Lancet 2009; 373(9661): 408-15, doi:10.1016/S0140-6736(08)61817-6 (in addition to having experienced discrimination,
experienced stigma, patients with mental problems sometimes also feel being discriminated
against, even when no discrimination has occurred, anticipated stigma). Moreover, at lines 308, 319, 328: about the stigma and the difficulty of open-self disclosure, especially in an
Emergency Department setting: “Suicidality Assessment of the Elderly With Physical Illness in the Emergency Department” Front Psychiatry 2020; 11:558974. doi:
10.3389/fpsyt.2020.558974.
Stigma, including, self-stigma or perceived stigma emerged as a key element and theme. These additional references have been most helpful. In particular the 2019 article ‘The Health Stigma and Discrimination Framework: a global, crosscutting framework to inform research, intervention development, and policy on health-related stigmas’ and the 2009 article ‘Global pattern of experienced and anticipated discrimination against people with schizophrenia: a cross-sectional survey’. While these are more general and not related to suicide specifically, they do provide comprehensive models of (mental) health related stigma. We have now included these in our discussion of the stigma-related findings found in this study.

2) About psychosocial factors, including the financial stressors: “Modelling suicide and
unemployment: a longitudinal analysis covering 63 countries, 2000-11” Lancet
Psychiatry 2015; 2(3): 239-45; “Impact of 2008 global economic crisis on suicide: time
trend study in 54 countries” BMJ 2013; 347: f5239;

“Economic suicides in the Great Recession in Europe and North America” Br J Psychiatry 2014; 205(3): 246-247; “Relationship of suicide rates to economic variables in Europe: 2000-2011” Br J Psychiatry 2014; 205(6): 486-496. Erratum in: Br J Psychiatry 2015; 206(2): 169; “Economic recession and suicidal behaviour: Possible mechanisms and ameliorating factors”. Int J Soc Psychiatry 2015; 61(1): 73-81.

Once again, these are most interesting and helpful references. We have now elaborated on our discussion of economic factors and their associations with suicide and have used a selection of the suggested references in making this point.

3) About relationships stigma-hopelessness-suicide (line 310) and, therefore, the
importance of an unconditional listening by health workers: “When Sick Brain and
Hopelessness Meet: Some Aspects of Suicidality in the Neurological Patient” CNS
Neurol Disord Drug Targets 2020;19(4):257-263, doi:
10.2174/1871527319666200611130804; “Neurological Diseases and Suicide: from
Neurobiology to Hopelessness” [Maladies neurologiques et suicide: de la neurobiologie
au manque d'espoir]. Rev Med Suisse 2015; Feb 11;11(461):402-5. PMID: 25895218.

Unfortunately, we could not access the full-text of the second reference in English. However, the first reference provided was very interesting and does provide a rich description of factors such as hopelessness and stigma in relation to suicidality in ‘the neurological patient’, in particular it was very interesting reading about the hopelessness experienced by these individuals in relation to a loss of functioning and in response to a lack of intervention/treatment. We have considered this reference carefully and have decided that it does not necessarily enhance our point regarding stigma between health professionals and those experiencing a recent suicide attempt in the absence of organic neurological conditions.

4) Interestingly, the “Interpersonal Trust” concept has been reported to be relevant to
virtually every facet of social functioning and has profound effects on mental and
physical health through an entire lifespan and under the most disparate conditions,
including suicide. For two cultural framings of this concept: Borum R “The Science of
Interpersonal Trust” Mental Health Law & Policy Faculty Publications 2010; 574,
http://scholarcommons.usf.edu/mhlp_facpub/574; “The conceptualization of
interpersonal trust: A basis, domain, and target framework” 2010. In: Interpersonal trust
during childhood and adolescence. Rotenberg KJ Ed, Cambridge University Press, New
York. Noteworthy, in the studies of Economu and colleagues during the Greek
economic crisis, the “Interpersonal Trust” emerged as the only significant protective
factor with respect to suicidal ideation: “Suicidal ideation and reported suicide attempts
in Greece during the economic crisis” World Psychiatry 2013; 12(1): 53-59; “Suicidal
ideation and suicide attempts in Greece during the economic crisis: an update” World
Psychiatry 2016; 15(1): 83-84

These are once again very interesting references. In our discussion of the importance of trust, rapport, and therapeutic alliance we now also note that interpersonal trust and social capital as described in these references is an important protective factor for suicidal ideation and provide an example reference from those provided.

- The Limitation section is present.
Thank you.

- The Conclusion rightly emphasizes the very important and pragmatic practical implication.
Best regards.

Thank you.

Round 2

Reviewer 2 Report

I remain stable on my first revision. The article should not be published in this journal, for the main concerns provided on my first report.

Author Response

We have noted the reviewer's comment in our cover letter to the editor.